# Mediterranean diet and associations with the gut microbiota and pediatric-onset multiple sclerosis using trivariate analysis
Ali I. Mirza [1,2], Feng Zhu[1], Natalie Knox[3,4], Lucinda J. Black[5,6,7], Alison Daly [5], Christine Bonner[3], Gary Van Domselaar[3,4], Charles N. Bernstein [8,9], Ruth Ann Marrie[8,10], Janace Hart[11], E. Ann Yeh[12], Amit Bar-Or [13], Julia O'Mahony[12], Yinshan Zhao[1], William Hsiao[2], Brenda Banwell[14], Emmanuelle Waubant[11] & Helen Tremlett [1] ✉

## Abstract

**Background** The interplay between diet and the gut microbiota in multiple sclerosis (MS) is poorly understood. We aimed to assess the interrelationship between diet, the gut microbiota, and MS.

**Methods** We conducted a case-control study including 95 participants (44 pediatric-onset MS cases, 51 unaffected controls) enrolled from the Canadian Pediatric Demyelinating Disease Network study. All had completed a food frequency questionnaire ≤21-years of age, and 59 also provided a stool sample.

**Results** Here we show that a 1-point increase in a Mediterranean diet score is associated with 37% reduced MS odds (95%CI: 10%–53%). Higher fiber and iron intakes are also associated with reduced MS odds. Diet, not MS, explains inter-individual gut microbiota variation. Several gut microbes abundances are associated with both the Mediterranean diet score and having MS, and these microbes are potential mediators of the protective associations of a healthier diet.

**Conclusions** Our findings suggest that the potential interaction between diet and the gut microbiota is relevant in MS.

## Plain language summary

Multiple sclerosis (MS) is a disease where the immune system attacks the protective covering of nerve cells in the brain. There may be a relationship between diet and bacteria within the gut and MS, however this is not well understood. We investigated how diet and gut bacteria are linked to MS in young people. We examined the diet and types of bacteria in stool samples from those with and without MS. We found that a diet richer in fiber and Mediterranean foods were less common in those with MS. This dietary pattern was linked to certain differences in the gut bacteria. These findings raise the possibility, but cannot prove, that what we eat may help prevent MS by influencing our gut bacteria. This research opens the door to further studies on how diet can impact MS through our gut bacteria.

While environmental risk factors for multiple sclerosis (MS) have been extensively studied, the most strongly associated determinants, such as low vitamin D status and prior infection with Epstein-Barr Virus, are not sufficient to cause MS on their own. It is likely they depend on other factors[1,2]. Modifiable factors such as diet and the gut microbiota are of particular interest as they can exhibit immunomodulatory and neuroprotective effects of relevance in MS[3,4]. The Mediterranean diet has been consistently associated with better health outcomes in the general population, including a reduced risk of mortality and lower risk of developing several chronic diseases, including some neurodegenerative conditions[5]. A pilot study suggested that a Mediterranean diet may alleviate MS symptoms, such as fatigue[6], but whether the diet is associated with a reduced risk of MS is unclear[7,8]. Studies evaluating early life diet are required to determine whether populations consuming a Mediterranean diet have a lower risk of MS. A few observational studies have found a potential relationship between specific components of the diet, including higher intakes of fruits, vegetables, and fish, as well as some nutrients (e.g., iron) and lower MS risk[9–12]. Combined, these early findings are encouraging, with the suggestion that aspects of diet could alter MS susceptibility.

Emerging evidence suggests that a perturbed gut microbiota may be linked with MS susceptibility[3]. For example, animal models have demonstrated that a transplanted stool sample from a person with MS can exacerbate, as well as lower the threshold of inducing, inflammatory CNS demyelination[13,14]. Key differences in the abundance of various gut bacteria and archaea, such as *Methanobrevibacter*, have also consistently been observed between individuals with and without MS[15–19].

---

The relationship between diet and gut microbiota is of growing interest as the microbiota may facilitate several health benefits[3,4]. Interestingly, the gut microbiota's gene composition (metagenome) in youth with pediatric-onset MS reflects a diet depleted in fiber, whole-grains, and iron[15]. However, the interplay between diet and the microbiota in people with MS is largely unexplored, especially in younger individuals early in their disease course[3]. Children and youth with MS have fewer exposures acting across their life course compared to adults, which may allow for more insight into the role of environmental factors in MS.

Here, we investigated the trivariate relationship between diet, the gut microbiota and pediatric-onset MS (the main outcome) in a case-control study. Our specific measures of diet included the Mediterranean diet score and components of that diet, as well as intakes of fiber, whole-grains, and iron. In addition, we also tested for microbiota taxa that may mediate the association of dietary intake with MS susceptibility. We observe that a higher Mediterranean diet score and higher nutrient intakes such as fiber, are associated with a lower odds of having MS, and that the gut microbiota might mediate this potential protective relationship.

## Methods

### Participant selection and data sources

Participant enrolment, data sources and procurement of stool samples and DNA sequencing have been described in detail elsewhere[15,19]. Briefly, MS cases and controls were enrolled through the Canadian Pediatric Demyelinating Disease Network study, which included participants from seven sites—six Canadian and one USA. All MS participants were <18 years old at the time of MS symptom onset, met the 2017 McDonald criteria and, when testing was available, were negative for serum anti–myelin oligodendrocyte glycoprotein antibodies using live, cell-based methods (Oxford, UK)[20,21]. Unaffected controls had no known neurological or immune-related conditions, although conditions such as headache/migraine, asthma and allergies were permissible. Informed consent was obtained from all participants and/or their parent/legal guardian. The University of British Columbia (UBC) Clinical Research Ethics Board (number: H15-03330) and the Hospital for Sick Children's (SickKids') Research Ethics Board (number: 1000051098) approved the study.

### The Diet group

Enrolled participants who had completed a Block Kids Food Screener (BKFS) before reaching 22 years of age were potentially eligible for the current study and formed the 'Diet group.' The BKFS is a validated 41-item food frequency questionnaire (FFQ) developed by NutritionQuest (Berkeley, CA, USA) to evaluate the habitual dietary intake of foods and nutrients in children and adolescents[9]. Participants or their caregivers were asked to recall the frequency and quantity of foods and beverages consumed by the participant during the previous 7 days. Participants with 15 or more missing answers on the FFQ or with implausibly low or high total energy intake (<500 or >3500 kcal/day) were excluded[22].

### Assessment of dietary intake and the Mediterranean diet scores

From the FFQ, the average daily intakes of food groups (grams/day) were estimated and were used to derive the widely used alternative Mediterranean diet score (aMED); considered an improved scale from the original with modifications to the dietary components based on associations with lower chronic disease risk and validated against blood biomarkers of inflammation and endothelial dysfunction[23]. We calculated the aMED scores based on sex-specific median intake from the controls for each of seven of the nine original dietary components, i.e., fruit, vegetables, legumes, whole-grain foods, fish, processed and red meat combined, and the ratio of monounsaturated fatty acids to saturated fatty acids (MUFA:SFA). Nuts and ethanol components were excluded as the FFQ did not capture them. One point was assigned if intake was >median, or, for processed and red meat only, if <median, and no points if =median or criteria not met (see Supplementary Table 1). Total possible scores for aMED ranged from 0–7, indicating the lowest to highest resemblance to the Mediterranean diet.

### The Diet-microbiota subgroup

The 'Diet-microbiota subgroup' included only those participants from the Diet group who provided a stool sample within +/−90 days of completing the FFQ (representing a time period during which major dietary patterns were observably relatively stable[24,25]), and who had not taken antibiotics or corticosteroids before 30 days of stool collection. If a participant had more than one available stool sample and FFQ, then the earliest sample procured and FFQ were included in the analyses.

### Participant characteristics

Participant characteristics, captured via questionnaires or standardized forms completed by site investigators and/or participants, included: sex, self-reported race (White, non-White) and the following characteristics at the time of the FFQ completion: age- and sex-standardized body mass index (BMI) percentiles (and overweight/obese categories)[26,27], history of comorbidities, and cigarette smoking (ever, never). At the time of stool sample procurement, participants completed a 7-point Bristol stool scale, which was categorized as hard (types 1–2), medium (types 3–5), or loose (types 6–7). Site investigators captured age at MS symptom onset, disease duration, and disease-modifying drug exposure status (ever or naïve) for the MS cases (Table 1). Further details on these characteristics and their categorization are in Supplementary Table 2.

### Stool sample collection, sequencing, and bioinformatics

Participants' stool samples were collected using a provided collection kit, and these were shipped with ice to the research laboratories and stored in a −80 °C freezer. DNA was extracted from stool samples using the Zymo Quick-DNA™ Fecal/Soil Microbe Miniprep Kit. The 16 S rRNA gene V4 region was amplified and sequenced at both ends up to 300 bp by the Illumina MiSeq platform. The paired-end reads were trimmed to 252 bp, merged, and denoised into amplicon sequence variants (ASVs) using Deblur (v.1.1.0), all via the QIIME2 (Quantitative Insights Into Microbial Ecology; v.2019.4) platform. We curated putative erroneous ASVs using LULU (v. 0.1.0)[28]. See Supplementary Methods for further details.

### Statistical analyses

For the Diet group, we assessed the association between individual dietary measures, which served as explanatory variables, including the aMED dietary pattern scores (0–7), nutrient intakes: whole-grains (oz equiv./day), fiber (g/day), and iron (mg/day) with MS, i.e., the primary outcome variable, categorized as "MS" or "control," all using logistic regression adjusted for age at time of FFQ completion, sex and total energy intake. Whole-grains (oz equiv./day) was calculated by determining the whole-grain component of each food item from the FFQ.

### Diet-microbiota subgroup analyses

For all downstream analyses, the microbial taxon counts were centered log-ratio transformed by taking the logarithm of the ratio of each taxon to the geometric mean of all taxa from the same sample, with a pseudo-count of 0.5 added to avoid computing the logarithm of zeros[29]. Only those genera/ASVs with a minimum relative abundance of 0.01% in at least 10% of all samples were further assessed. We assessed the association between the gut microbiota taxa, which served as the explanatory variables, and the odds of MS, i.e., the outcome variable. All taxa were standardized to Z-scores and included the relative abundance of 201 genera and 770 ASVs, as well as the presence of the genus *Methanobrevibacter*. We did this using logistic regression adjusting for library size, measured as total abundance of species sequenced from an individual's stool sample, as well as the Bristol Stool Scale group, age and sex.

We also examined the associations between dietary measures, which served as the explanatory variables, and each taxon, i.e., the outcome variable in this case. In this specific analysis, we standardized only the dietary measures to Z-scores. These included the aMED score, the aMED seven dietary components (g/day), whole-grains (oz equiv./day), fiber (g/day), and iron (mg/day) intake. We used linear regression, additionally adjusted for

**Table 1 | Characteristics of the pediatric-onset MS cases and controls at the time of food frequency questionnaire completion (unless otherwise stated)**

| Characteristic | MS cases, n = 44 | Controls, n = 51 |
|---|---|---|
| Sex, female: No. (%) | 33 (75%) | 32 (63%) |
| Self-identified race: White[a], No. (%) | 14 (32%) | 12 (24%) |
| Age, years: median (Q1, Q3) | 18 (16, 19) | 16 (15, 18) |
| Age- and sex-specific BMI percentiles[b]: median (Q1, Q3) | 71 (48, 85) | 54 (29, 81) |
| Overweight/obese (≥85th percentiles)[b]: No. (%) | 8 (18%) | 7 (14%) |
| Atopy-related conditions (asthma, dermatitis, psoriasis, and acne): No. (%) | 8 (18%) | 7 (14%) |
| Other comorbidities[c]: No. (%) | 3 (7%) | 1 (2%) |
| Cigarette smoking (ever)[d]: No. (%) | 2 (5%) | 0 (0%) |
| **Dietary intake** | | |
| Mean (SD): | | |
| aMED score[e] | 2.6 (1.4) | 3.4 (1.4) |
| Median (Q1, Q3): | | |
| Total energy (kcal/day) | 1100 (800, 1400) | 1100 (830, 1400) |
| Whole-grains (oz equiv./day) | 0.33 (0.11, 0.56) | 0.48 (0.26, 0.92) |
| Fiber (g/day) | 9.4 (5.7, 12) | 11 (7.9, 15) |
| Iron (mg/day) | 7.2 (5.6, 10) | 8.7 (6.2, 11) |
| **MS specific** | | |
| Age at symptom onset, years: median (Q1, Q3) | 16 (14, 17) | – |
| Disease duration, years: median (Q1, Q3) | 0.81 (0.43, 3.9) | – |
| Disease-modifying drug exposure status (ever/naïve)[f]: No. (%) | 26 (59%)/ 18 (41%) | – |
| Ever beta-interferon | 13 (30%) | – |
| Ever glatiramer acetate | 8 (18%) | – |
| Ever dimethyl fumarate | 3 (7%) | – |
| Other[g] | 2 (5%) | – |
| **Subgroup only: Microbiota-related metric** | **MS cases, n = 27** | **Controls, n = 32** |
| Bristol Stool Scale[h]: median (range) | 3 (2–6) | 4 (1–7) |
| Hard (types 1–2): No. (%) | 6 (22%) | 6 (19%) |
| Medium (types 3–5): No. (%) | 18 (67%) | 25 (78%) |
| Loose (types 6–7): No. (%) | 3 (11%) | 1 (3%) |

MS cases (n = 44) and controls (n = 51) differed by age (P = 0.011, Wilcoxon rank-sum test). Other than dietary intake, no other characteristics differed (P > 0.16). The median (Q1, Q2) number of the absolute days between these two time points for all participants was 6.0 (3.0, 15).
[a]The number of MS/controls missing race data was 18/21.
[b]The number of MS/controls missing body mass index (BMI) data was 14/20.
[c]Other comorbidities were: attention deficit hyperactivity disorder (n = 1 MS), holoprosencephaly (n = 1 MS), hypothalamic/pituitary dysfunction (n = 1 MS), and hypothyroidism (n = 1 control).
[d]Includes passive and active smokers.
[e]For the aMED, no participant scored 7 (representing the highest resemblance to a Mediterranean diet) while 1 MS case (and no controls) scored 0.
[f]Ever/naïve refers to ever or never use of a disease-modifying drug. The number of MS cases ever a disease-modifying drug or never using was the same whether assessed at the time of pre-FFQ completion or pre-stool sample collection.
[g]Other disease-modifying drugs at the time pre-FFQ were: rituximab (n = 1) and natalizumab (n = 1).
[h]The Bristol stool types were recorded at the time of stool sample collection.

total energy intake. To prevent influence from outliers, all regression analyses involving taxa abundances mentioned earlier were performed using generalized linear regressions based on Mallow's type robust quasi-likelihood estimator via the R package robustbase (v. 0.93-9, method equal to"Mqle," default settings)[30,31].

Next, we examined the possible role of the microbiota to mediate the association between diet—specifically the aMED score and fiber intake—and MS odds by conducting a mediation analysis within the potential outcomes framework via the R package mediation, v. 4.5.0[32,33]. We chose to evaluate fiber intake alongside the aMED score given its established influence on gut microbiota composition[3]. In the Diet-microbiota subgroup, the abundances of gut microbiota taxa were individually tested as candidate mediators. This was only performed for taxa that exhibited significant associations (P < 0.05) with the aMED or fiber, and with odds of MS. The indirect effect, within this framework, represents the association between diet and MS odds, i.e., the outcome variable, through the mediating variable, namely an individual gut microbiota taxa. It quantifies the difference in the odds of MS when comparing the mediator variable values of 0 and 1, while holding the dietary value constant. Conversely, the direct effect represents the unmediated association of diet on the odds of MS, and quantifies the difference in odds when comparing the dietary values of 0 and 1, while holding the mediator variable constant. These two effects sum up to the total effect[32]. The odds ratios and their 95% confidence intervals were calculated using the Quasi-Bayesian approximation with 10,000 Monte Carlo draws. Mediation was hypothesized when the indirect effect reached a significance level of P < 0.05.

Results were reported as adjusted odds ratios (ORs) or beta coefficients, as appropriate, with 95% confidence intervals (CI) and two-sided P-values. The correlations, detailed in the complementary analysis, and beta coefficients were reported as having a strong association if the absolute value was ≥0.70 (arbitrarily assigned). High-dimensional tests were adjusted for multiple comparisons using the Benjamini–Hochberg method and reported as Q-values. All statistical analyses were conducted using R, v. 4.0.4[34].

### Complementary analyses

We conducted complementary analyses, including, for the Diet group only: assessing the association between the seven aMED dietary component intakes (g/day) with MS. The component intakes were also dichotomized to binary scores of 0 or 1 based on the sex-specific median intakes of controls, see Supplementary Table 1, and their associations were also assessed. We also assessed the interactions of age and sex with the associations between aMED scores or nutrients and MS odds. We also examined the pairwise associations between the dietary pattern scores, their individual dietary components intakes and scores, and nutrients, using Spearman's correlation coefficient (ρ). We also adjusted the main diet-related findings for over-weight/obese status, as well as self-reported race, recognizing that race is a social construct and structural racism has adverse outcomes on many health indicators[35].

For the Diet-microbiota subgroup only, we investigated the sources of inter-individual variation in the gut microbiota composition and then assessed the relationship of the diet-associated variation of the microbiota with MS risk using multivariate analyses. The details are in the Supplementary Methods.

We conducted additional analyses: first, we compared the relative abundance of microbial taxa by vitamin D supplementation and disease-modifying drug exposure status among MS cases using the Wilcoxon rank sum test; second, we re-examined the associations between dietary measures and each taxon among those participants who completed the FFQ within 6 days of stool sample collection, using linear regression.

### Reporting summary

Further information on research design is available in the Nature Portfolio Reporting Summary linked to this article.

### Results

Ninety-five participants fulfilled our inclusion criteria for the Diet group, consisting of 44 pediatric-onset MS cases and 51 controls. Among these, 59 were included in the Diet-microbiota subgroup, comprising 27 pediatric-onset MS cases and 32 controls. A flowchart depicting the participant selection process is provided in Supplementary Fig. 1. There were no

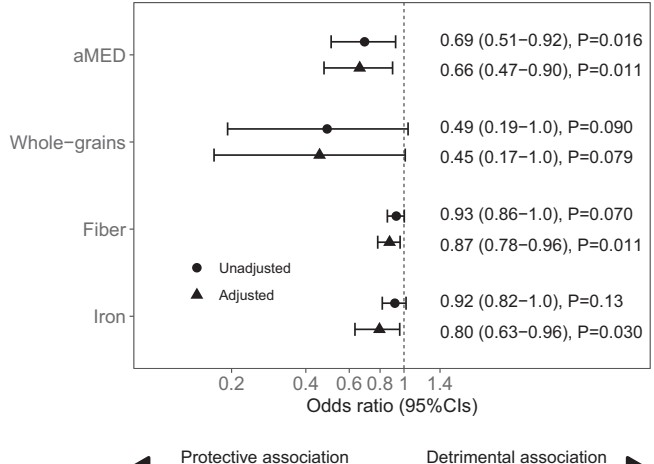

**Fig. 1 | The association between the Mediterranean diet score or nutrient intake and odds of MS.** aMED alternate Mediterranean diet score, OR odds ratio, CIs confidence intervals. The associations between the aMED score, whole-grains (oz equiv./d), fiber (g/d), and iron (mg/d) intakes and multiple sclerosis (MS) risk are represented as odds ratios adjusted for total energy intake (kcal/d), age (in years) at food frequency questionnaire completion, and sex, their 95% CIs, and P-values. A total of n = 44 MS cases and n = 51 controls were assessed. Results were similar when reassessed for the subgroup of participants with only a stool sample (n = 27 MS cases and n = 32 controls), albeit associations with iron did not reach significance (P > 0.05). The x-axes units are in log 10 scale. An OR < 1 is a "protective" association, meaning that an increase in the dietary intake is associated with a lower odds of MS, and an OR > 1 is a 'detrimental' association, meaning that an increase in the dietary intake is associated with a higher odds of MS. For example, an OR of 0.66 for aMED means that a 1-point increase in the aMED score for an individual is associated with a 34% lower odds of MS. Source data are provided as a Source Data File.

significant differences in the characteristics of the MS cases and controls, as shown in Table 1, all P < 0.050. However, MS cases were, on average, two-years older than controls at the time of completion of the food frequency questionnaire (FFQ) (Wilcoxon rank-sum test, P < 0.037). Females predominated, representing 75% of MS cases and 63% of controls. All MS participants had a relapsing-remitting disease course. Of the 44 MS cases, 41% (18) were disease-modifying drug naïve. The median disease duration from MS symptom onset to completion of the FFQ was <1 year (Table 1).

### Diet-only related results
A higher alternative Mediterranean diet score (aMED) score and higher intakes of fiber (g/day), whole-grains (oz equiv./day), and iron (mg/day) were each associated with a lower likelihood of having MS, albeit whole-grains did not reach significance (Fig. 1, Supplementary Fig. 2). For example, a 1-point increase in the aMED score was associated with 34% lower odds of MS (adjusted odds ratio [$OR_{adj}$] 0.66, 95% CI 0.47–0.90, P = 0.011), and a 1-gram increase in fiber was associated with 13% lower odds ($OR_{adj}$ 0.87, 95% CI 0.78–0.96, P = 0.011).

### Diet-microbiota related results
In the Diet-microbiota subgroup, most participants, 53 out of 59, completed their FFQ within a shorter 30-day period relative to stool collection, with the overall median interval just 6 days between FFQ completion and stool sampling. When comparing disease status, MS cases vs controls, significant differences were found for 7 genera and 12 amplicon sequence variants (ASVs), all P < 0.050, although all Q > 0.68, see Supplementary Data 1. ASVs are taxonomic units at the species-level. The relative abundances of all taxa were standardized to Z-scores and served as the explanatory variable. Among the genera examined, *Methanobrevibacter* abundance was the most enriched in MS cases relative to controls and a 1-standard deviation increase in its relative abundance was associated with a 5-fold higher odds of MS, $OR_{adj}$ 5.7, P = 0.032. Moreover, *Methanobrevibacter* was detected in 33% of

MS cases (9/27) compared to a just 3.1% of controls (1/32). Its presence corresponded to a 23-fold higher odds of MS, albeit confidence intervals were wide, $OR_{adj}$ 23, 95%CI 3.0–540, P = 0.011, logistic regression. The prevalence of all other genera and species within each group, i.e., the MS cases vs controls, can be found in Supplementary Data 1. In contrast, *Ruminococcaceae NK4A214 group* genus was the most depleted in MS and a 1-standard deviation increase in its relative abundance was associated with a 68% lower odds of MS ($OR_{adj}$ 0.32, P = 0.0034). Several taxa that reached statistical significance at the ASV-level were also significant at the genus-level findings, including ASVs housed in *Variovorax*, *Eggerthella*, *Lactococcus* and *Ruminococcaceae NK4A214 group*.

Dietary measures, serving as standardized explanatory variables, exhibited a strong relationship with 3 of the 7 (43%) genera and 3 of the 12 (25%) ASVs that were identified as statistically different by disease status, i.e., differed between the MS participants and controls. These dietary measures included intakes of fruits, the ratio of monounsaturated fatty acids to saturated fatty acids (MUFA:SFA), whole-grains, fiber, and iron. The absolute beta coefficients, $β_{adj}$, were ≥0.70, as shown in Fig. 2, Supplementary Data 2. The aMED score was inversely associated with the following taxa identified to be more abundant in the MS participants relative to the controls: *Methanobrevibacter*, *Eggerthella sp.* and *Lactococcus sp.*, with $β_{adj}$ ranging from −0.34 to −0.52, P < 0.019. The aMED score was positively associated with the following taxa which also had lower abundance in the MS participants relative to controls: *Clostridiales vadin BB60 group* and *Ruminococcaceae NK4A214 group sp.* ($β_{adj}$ > 0.32, P < 0.025), the latter being most strongly associated with iron intake ($β_{adj}$ 1.6, P = 2.9 × 10^{-13}). Compared to the aMED, fiber intake was more strongly associated with all of the aforementioned taxa (Fig. 2, Supplementary Data 2).

A total of 4 genera and 6 ASVs, were associated with both diet (the aMED or fiber intake) and disease status (Fig. 1, Supplementary Data 2); therefore, we further tested these microbiota measures as candidate mediators of the association of aMED (and fiber intake) with MS risk. In total, seven taxa were significant potential mediators (P-values of the indirect [mediated] effect <0.045, Fig. 3), of which, *Methanobrevibacter* was the strongest mediator by proportion mediated. Interestingly, *Anaerovoracaceae Family XIII AD3011 group sp.* was a significant competitive mediator meaning that it may partially offset the protective association of aMED and fiber intake with MS risk.

### Complementary analyses results
The complementary analyses did not change our interpretation of findings. For the Diet group, although most participants reported consuming few to no legumes, an increase in the intake of this component by one-standard deviation was associated with 55% lower odds of MS ($OR_{adj}$ 0.45, 95%CI 0.18–0.85, P = 0.041). While the median daily intakes in grams were lower in MS cases, relative to controls, for fruits, vegetables, and whole-grain foods, and higher in processed and red meat, none reached statistical significance, all P > 0.17, see Supplementary Table 3a. However, only 25% of MS cases scored a point for whole-grain foods compared to 47% of controls ($OR_{adj}$ 0.40, 95% CI 0.16–0.97, P = 0.047, Supplementary Table 3b).

Age and sex did not significantly interact with any dietary measures in logistic regression models of MS risk (P > 0.067). As expected, fiber intake strongly correlates with the aMED score and fruit intake (both ρ = 0.71, Supplementary Fig. 3b). Adjusting for race and overweight/obese status strengthened the protective associations of the aMED and nutrients with MS risk, as shown in Supplementary Table 4. For example, a 1-unit increase in the aMED score was associated with an additional 5% (i.e., 0.05) lower odds of MS after additional adjustments ($OR_{adj}$ 0.61, 95% CI 0.43–0.84, P = 0.0040), as was consumption of whole-grains ($OR_{adj}$ 0.40 95%CI 0.14–0.94, P = 0.058).

For the Diet-microbiota subgroup, a trivariate relationship between MS, the overall gut microbiota and diet was identified in a stepwise approach, shown in Supplementary Figs. 4 and 5. When comparing sources of inter-individual variation in the overall gut microbiota at the genus-level, race explained the most (5.7%), followed by fruit and then fiber intakes and

**Fig. 2 | The Diet-microbiota subgroup: heatmap of the relationship between dietary measures, microbial taxa, and disease status.** MS multiple sclerosis, aMED alternate Mediterranean diet score, P&R meat processed and red meat, MUFA:SFA ratio of monounsaturated to saturated fat, sp. unknown species. **a** displays the 7 genera and **b** displays the 12 ASVs that were significantly associated with MS (*P* < 0.050). The association of each dietary measure Z-score with taxa relative abundance (the outcome variable) was represented as beta coefficients. The association of each taxa Z-score with disease status (*n* = 27 MS cases vs *n* = 32 controls, the outcome variable in this case) was represented as log odds ratios (far most right column). All results were adjusted for library size, Bristol Stool Scale group, age (in years) at the food frequency questionnaire completion and sex while beta coefficients were also adjusted for total energy intake. Dietary measures and taxa were standardized to Z-scores only when modeled as an independent variable. Actual estimates and associated *P* and Q values are available in Supplementary Table. Heatmap was created using the R package pheatmap (v. 1.0.12). Source data are provided as a Source Data File.

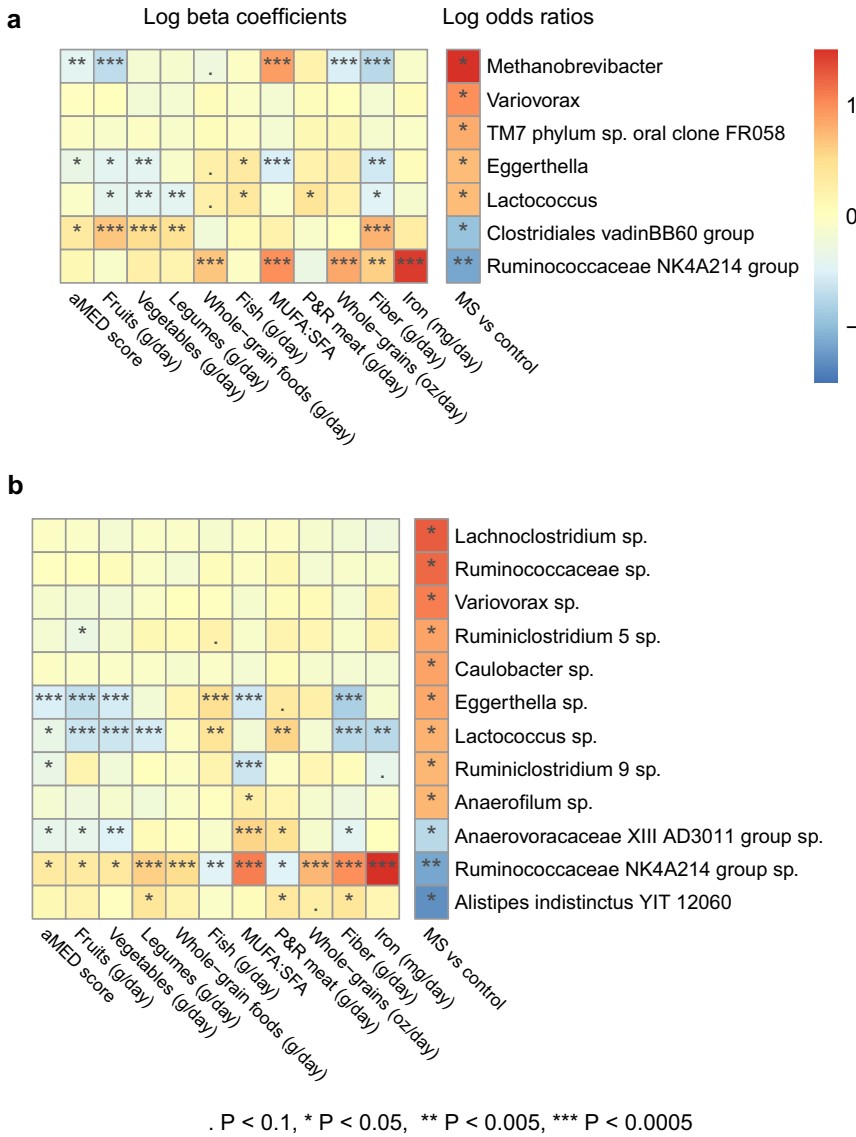

. P < 0.1, * P < 0.05, ** P < 0.005, *** P < 0.0005

then the aMED score (3.0%–4.3%, all *P* < 0.0050, PERMANOVA), while MS itself, relative to controls, explained only 1.7%, *P* = 0.41, see Supplementary Fig. 4a. We generated constrained principal components (PC) that represented the inter-individual variance of the gut microbiota composition explained (that is, constrained) by the aMED or fiber intake, defined as PC-aMED and PC-Fiber, respectively. The PC-aMED or PC-fiber scores were associated with lower odds of MS (OR$_{adj}$ 0.41–0.45, *P* < 0.020, Supplementary Fig. 4b, c). The results at the ASV-level were consistent with those at the genus-level and support the same interpretation, see Supplementary Fig. 5.

Our analysis found no evidence that the association of specific microbial taxa with MS and dietary items was explained by either vitamin D supplementation or disease-modifying drug exposure status (Supplementary Tables 5 and 6).

In total, 32 participants (10 MS cases and 22 controls) had completed the FFQ within 6 days of stool sample collection. Our findings were largely consistent with the original analysis. For example, fiber was strongly associated with *Eggerthella*, *Lactococcus*, *Clostridiales vadin BB60 group* and *Ruminococcaceae NK4A214 group* (the absolute beta coefficients, β$_{adj}$, were ≥0.90, *P* < 0.00050). However, *Methanobrevibacter* was no longer associated with aMED or fiber intake (*P* > 0.05), likely attributable to its limited prevalence as it was detected in only 3 of the 32 remaining participants, all of which had MS.

## Discussion

A higher Mediterranean diet score and higher intakes of specific nutrients were associated with a lower likelihood of having pediatric-onset MS. We demonstrated that the gut microbiota might facilitate this protective association. Key findings included that a higher alternative Mediterranean diet score (aMED) and higher intakes of fiber and iron were each associated with a lower likelihood of having pediatric-onset MS. Consistent with previous observations, relative to controls, MS cases were enriched with the archaeon *Methanobrevibacter*[15,36–38]—a gut-dominant methane-producer associated with constipation[39], a common gastrointestinal complaint in individuals with MS[40]—and *Eggerthella*[41,42]. Intriguingly, the presence of *Methanobrevibacter* in our cohort was associated with a significantly higher likelihood of having MS, albeit the 95%CIs were wide (OR$_{adj}$ 23, 95%CI 3.0–540). Furthermore, our MS cases were depleted of *Clostridiales vadinBB60 group* and *Ruminococcaceae NK4A214 group*[19], the latter is a genus from a short-chain fatty acid (SCFAs)-producing family[43]. Immunomodulatory SCFAs are prominent bacterial products of fiber fermentation[44]. Our study is unique in that it links dietary measurements such as the aMED and dietary fiber to these MS-associated taxa. Notably, these particular taxa were significant potential mediators of the association of both the aMED and fiber intake with the risk of MS. Fiber intake and the aMED score were significant sources of inter-individual variation in the overall gut microbiota, while age, sex, and MS (vs controls) were not,

**Fig. 3 | The Diet-microbiota subgroup: forest plot of mediation analysis for diet, potential gut microbiota mediators, and MS odds.** aMED alternative Mediterranean diet score, sp. unknown species Participants assessed include *n* = 27 MS cases and *n* = 32 controls. The "indirect effect" (blue circle) is the association of the dietary measure with MS risk mediated by the microbiota. Mediation by the microbiota is hypothesized when the indirect effect is significant. The "direct effect" (yellow circle) is the non-mediated association of diet with MS risk. The "total effect" (gray circle) is the combined mediated (indirect effect) and non-mediated (direct effect) association of the dietary intake (aMED or fiber intake) with MS risk. Because the total effect is dependent on the mediator, the total effect changes when the mediator is different. Results are represented as odds ratio (OR) adjusted for total energy intake, library size, the Bristol Stool Scale group, age at the FFQ completion and sex. An OR < 1 is a "protective" association and an OR > 1 is a 'detrimental' association. Only PCs and taxa significantly associated with both diet (aMED or fiber) and MS risk were displayed and tested for mediation. All Q-values of the indirect effect were >0.10 for aMED and >0.069 for fiber. The circles and error bars represent the mean and 95% confidence intervals of the OR, respectively. Thick error bars represent a significant association with MS risk (*P* < 0.05). Source data are provided as a Source Data File.

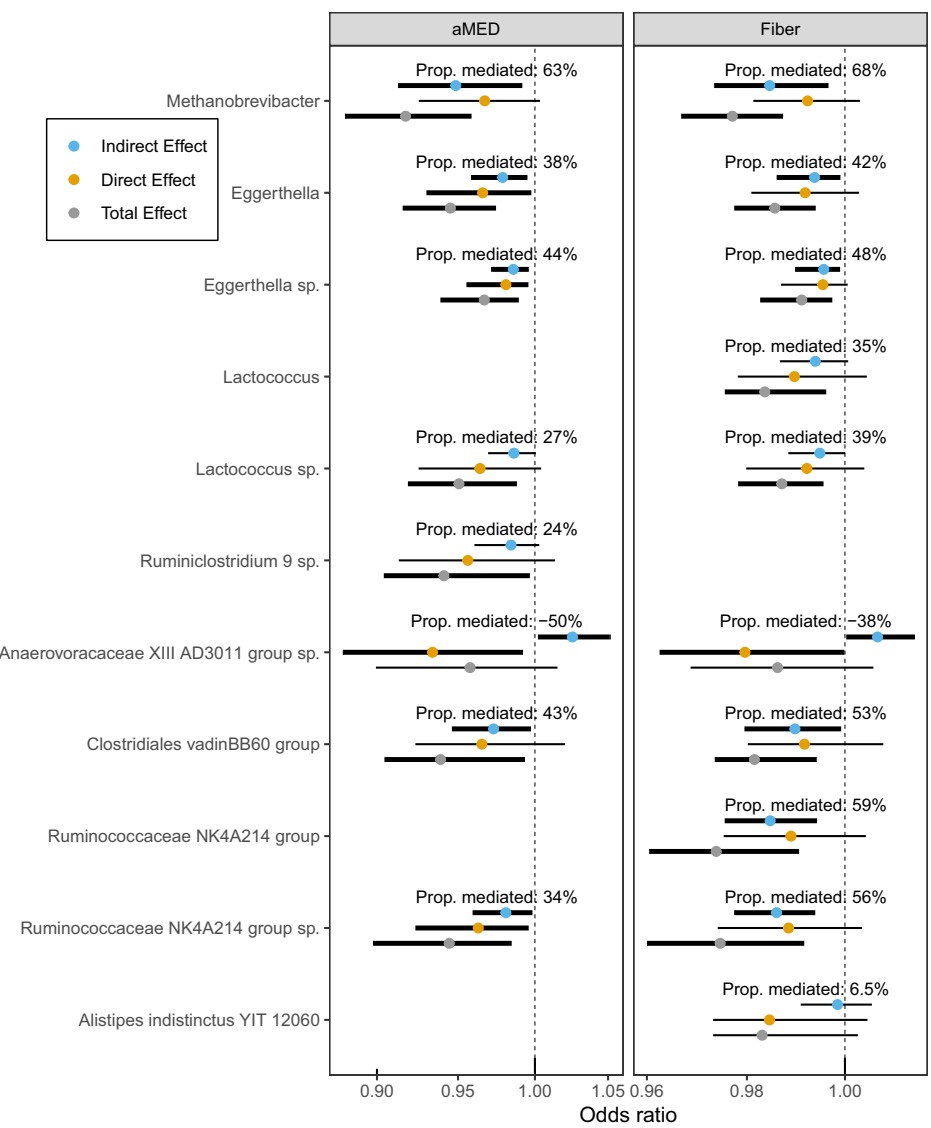

indicating that diet may have had an important impact on the gut microbiota composition. Remarkably, the specific aMED- and fiber-associated variations of the microbiota in turn were associated with lower risk of MS. Together, findings suggest the interplay between the host, microbiota and diet are likely relevant in MS.

We found just two other published studies investigating the Mediterranean diet with MS risk. Both studies involved adults exclusively and neither included the gut microbiota[7,8]. Consistent with our findings, a cross-sectional study from Iran compared 69 MS cases and 140 controls (82% women) and reported that the highest versus lowest tertile on the Mediterranean diet score was associated with a 77% reduced risk of MS risk[7]. However, a large longitudinal study of over 185,000 US female nurses, including 480 individuals with MS, did not observe an association between the aMED scores (assessed as quintiles) and the risk of developing MS[8]. This US-based study was the only one to include alcohol as a (healthy) component of the Mediterranean diet, while recent evidence suggests that it is not healthy[45]. Emerging evidence suggests that the Mediterranean diet may help modify MS course and related symptoms. A randomized controlled pilot trial and two cross-sectional studies reported a higher Mediterranean diet score among MS cases was associated with reduced disease severity, disability, fatigue and depression[6,46,47]. Together, these findings warrant further investigations into the potential health benefits of a Mediterranean diet for individuals with or at risk for MS.

Motivated by our prior research conducted on a considerably smaller cohort from the same source population[15,48], we assessed the nutritional intakes of fiber, whole-grains, and iron. We observed that consumption of these nutrients were lower in the diets of MS cases (relative to controls) and higher intakes were associated with reduced likelihood (odds) of having MS. Consistent with these findings, lower nutritional intakes of fiber and iron by MS cases compared to controls were observed in two studies: a US multi-center study of 312 pediatric-onset MS cases and 546 controls and an Israel-based study of 63 adult MS cases and 83 controls[9,49]. Similarly, an Iran-based study also observed a lower dietary fiber intake among 77 MS cases compared to 148 controls[50]. While no other study has assessed the intake of whole-grains as a food component in MS cases, the consumption of whole foods that are primarily composed of whole-grain—a healthy component of the aMED—has been previously evaluated. Although the direction of their findings was similar to ours, none reached significance, at least when adults were asked to recall their diet from ages 6 to 20 years[11]. We were able to show a significant relationship with a high intake of whole-grain foods associated with a 60% lower odds of MS. Interestingly, whole-grain food consumption was found to have an inverse relationship with MS disability in a cohort of participants enrolled in the North American Research Committee on MS (NARCOMS) study[51]. Combined, evidence suggests that a diet rich in fiber, whole-grains, and adequate iron—found in a Mediterranean diet—may be protective for MS onset.

Consistent with previous observations, the consumption of other healthy components of the aMED were also associated with a reduced risk of MS in our study, such as legumes[11,50], fruit[7,11], and vegetables[7,50], but only legumes reached significance. Surprisingly, MS participants from the International Multiple Sclerosis Microbiome Study (iMSMS) consumed more fruits and vegetables, and scored better overall on a healthy eating index, when compared with household controls, who were predominately their spouses[18]. These counterintuitive and seemingly contradictory findings may be attributed to reverse causation, as those affected by a disease such as MS may later decide to adopt a healthier diet consisting of fruits and vegetables[52]. Participants in the study averaged 50 years of age and had a long MS disease duration (averaging 14 years). Sex differences between cases and controls may also contribute, as most MS cases were women, while most household controls were men. Women are reported to consume more nutritious diets than men[53].

Diet is an important determinant of microbiota composition, and diet-induced shifts in gut microbiota composition have been linked to health benefits[54]. Findings from our correlation and mediation analyses indicate that dietary habits may explain differences between MS cases and controls for several gut taxa. These findings also concur with the dietary intervention studies in animals and humans. For example, animals diets high in fiber such as resistant starch and grains were associated with a reduced abundance of *Eggerthella*[55], and *Methanobrevibacter*[56,57]. These genera were also enriched in our MS cases. Interestingly, reduced carbohydrate and fiber consumption rapidly increased the abundances of *Eggerthella* and *Lactococcus*[58]. Similarly, a short-term, 1–3 day reduction of carbohydrate and fiber intake resulted in an increased abundance of *Eggerthella* and *Lactococcus* in a study of ten participants[58]. Studies in animals have shown that *Ruminococcaceae NK4A214 group abundance*—depleted in our MS cases—increased after a high grain diet in cows and decreased after a low iron diet in mice[59,60]. Interestingly, we observed that an ASV from the *R. NK4A214* bacterial group was associated most strongly with dietary iron followed by fiber intake. Also consistent with our findings, a high fat, high animal protein diet in rats was associated with an increased abundance of the *Anaerovoracaceae Family XIII AD3011 group*[61]. Intriguingly, our mediation analysis suggests that an ASV from the *A. Family XIII AD3011 group*, which was depleted in our MS cases (relative to controls), may partially offset the protective association of the aMED or fiber intake with MS risk. These findings warrant further investigation into the potential of the gut microbiome to mediate the effects of diet on MS risk.

Others have also identified putative inter-relationship(s) between diet, the gut microbiota and MS, but primarily using correlation and network analysis[3,18,36,62,63]. For example, a study linked meat servings with lower gut *Bacteroides thetaiotaomicron* abundance (a fiber digesting bacterium), higher circulating T-helper 17 cell and higher abundance of meat-associated blood metabolites in 185 MS cases relative to 330 controls[62]. Also, the iMSMS study observed an overlap between the gut microbiota taxa associated with MS and intakes from several food groups, especially fruits[18]. Although many of these studies are modest in size or provide indirect evidence, together with our present work, they suggest that diet-associated alterations of the gut microbiota may play a role in MS pathogenesis.

Pediatric-onset MS is a rare disease, with most studies reporting an estimated annual incidence of <1 per 100,0000 population. Hence although our study size was modest, it represents a valuable and unique cohort that could be highly informative[64,65]. Nonetheless, cases and controls were demographically well balanced and similar for stool consistency, an important confounder in gut microbiota studies and seldom reported in the MS literature[66]. Furthermore, our MS cases had a short disease duration, averaging <1 years from symptom onset (at time of FFQ completion), and thus would have incurred relatively few competing (or confounding) exposures, such as comorbidities, polypharmacy and other lifestyle changes that occur when living with chronic disease, especially when compared to adults with MS. Our study had limitations. First, the study was cross-sectional in design and data collection occurred after disease onset, and

therefore, a causal relationships between diet, the microbiota, and MS could not be determined. Second, guided by the literature, we allowed a +/− 90 day window between completion of the FFQ and stool sample procurement as substantial dietary changes are unlikely to have occurred during this time[24,25]. While it is possible that a shorter time window may be preferable, most participants (90%) completed their FFQ within a narrower 30-day period. Nonetheless, even when participants with the shorter window of +/− 6 days between the FFQ and stool sample collection were examined, findings remained largely consistent and a strong fiber-microbiota correlation remained. Third, we cannot exclude the possibility that individuals with MS changed their dietary habits after symptom onset or diagnosis, although, this is unlikely as disease duration (from symptom onset) was short. Fourth, we lacked information on nuts and alcohol intake and therefore could not compute the aMED score based on all of its original healthy components. However, alcohol use in our young cohort would be expected to be low[45]. Fifth, we did not have access to participants' physical activity levels, sunlight exposure or serum vitamin D levels which prevents us from evaluating their possible impact on the relationship between diet, the gut microbiota and MS risk. Nonetheless, to our knowledge, no study has demonstrated a link between physical activity and subsequent risk of developing pediatric-onset MS. Sixth, our modest cohort size limited our ability to draw robust conclusions, especially for specific microbial taxa, such as *Methanobrevibacter*, which was infrequently detected. This size limitation could reduce our ability to uncover true relationships. It also remains possible that some of our findings could have occurred by chance alone. These constraints collectively necessitate a cautious interpretation of the findings. Despite these limitations, this study provides valuable insight into the diet-microbiome-MS relationship. In particular, we identified key potential dietary and microbial risk factors for MS which generate intriguing hypotheses regarding their underlying mechanisms. Furthermore, based on our mediation analysis, we hypothesize that the potential protective effect of a fiber-rich nutritious diet against MS may be attributed to its potential influence on the gut microbiota. To help determine if this hypothesis reflects a causal relationship, conducting a large longitudinal study would be of value.

To conclude, a higher score on the alternative Mediterranean dietary metric, and higher intakes of fiber and iron, were significantly associated with a reduced likelihood of pediatric-onset MS. Furthermore, a protective relationship was identified between the aMED- and fiber-associated variation in the gut microbiota, in terms of both individual taxa abundance and overall composition, and MS risk. Combined, our findings suggest that an interaction between the host, the gut microbiota and a healthy diet rich in fiber such as the Mediterranean diet, is likely important in MS. Further work is warranted to characterize the diet-microbiome-MS trivariate relationship.

## Data availability

The sequencing data associated with this study are available under the National Center for Biotechnology Information (NCBI) BioProject accession number PRJNA1000059. Source data for all figures are available in Supplementary Data 3. Other data that support the findings of this study are available from the study team (contact the corresponding author, H.T.), upon reasonable request.

## Code availability

The microbiota bioinformatics were implemented in Qiime2 (v. 2019.4) are available for download at https://qiime2.org/. The statistical analyses were implemented in the R programming language (v. 4.0.4), available for download at https://www.r-project.org/. The relevant packages used for analyses were listed in the corresponding methods section in the article and supplemental file. The codes that support the findings of this study are available from the study team (contact the corresponding author, H.T.), upon reasonable request.

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

## Acknowledgements

The authors are grateful for all of the involvement of the participants, especially children and youth with multiple sclerosis, and their parents without whom this study would not have been possible. The authors also acknowledge the important contributions of: the Tremlett team (University of British Columbia); Dr. Morag Graham (National Microbiology Laboratory, Public Health Agency of Canada) in providing guidance for the microbiome sequencing; Thomas Duggan in facilitating study set-up, coordination and data collection; Bonnie Leung for study coordination; Michael Sargent (Department of Internal Medicine, and the University of Manitoba IBD Clinical and Research Center laboratory, Winnipeg, MB, Canada) for managing the stool biobank, and Dr. Jessica D. Forbes (University of Toronto, Toronto, Canada) for assisting with the original grant application. We are also grateful to the investigators and study teams at each site who participated in the Canadian Pediatric Demyelinating Disease Network study, including Dr. Douglas Arnold. Finally, our study would not have been possible without the support of the endMS Personnel Award from the Multiple Sclerosis Society of Canada.

## Author contributions

A.I.M. designed the research question with support from H.T.; A.I.M. and H.T. contributed to conception and design of the study, acquisition, interpretation of data, and drafted the first version of the manuscript, tables and figures. F.Z., Y.Z., and A.D. provided input on statistical analysis. L.B. and W.H. contributed to the study design. Y.Z., G.V.D., C.B., J.H., E.W., and H.T. facilitated obtaining funding (PI: Tremlett, The Multiple Sclerosis Scientific and Research Foundation). J.H. facilitated training of study coordinators the collection of stool samples. C.N.B. oversaw the biobanking. N.K. and G.V.D. oversaw the 16 S rRNA sequencing; C.B., performed the stool extractions and 16 S rRNA sequencing; A.I.M. conducted the bioinformatics and analyses. All authors—A.I.M., F.Z., L.B., A.D., C.B., G.V.D., C.N.B., R.A.M., J.H., E.A.Y., A.B.-O., J.O., Y.Z., W.H., B.B., E.W., H.T.—contributed to the interpretation of the data and provided a critical review of the manuscript.

## Competing interests

This study was supported by the Multiple Sclerosis Scientific and Research Foundation (#EGID: 2636; PI:Tremlett). The funding source was not involved in the study design, the collection, analysis, and interpretation of the data, or in the decision to submit this article for publication. A.I.M. is funded through the MS Society of Canada endMS Doctoral Studentship (EGID: 3246) and The Multiple Sclerosis Scientific and Research Foundation (PI: Tremlett, EGID: 2636). F.Z. and Y.Z. were funded through research grants held by H.T., including The Multiple Sclerosis Scientific and Research Foundation (PI: H.T., EGID: 2636). L.B. and A.D. are supported by Multiple Sclerosis Society of Western Australia (MSWA). L.B. is also supported by an MS Australia Postdoctoral Research Fellowship. D.L.A. is funded by the Canadian MS Society, the International Progressive MS Alliance, the Canadian Institutes of Health Research and the US Department of Defense. He has received personal compensation for serving as a Consultant for Alexion, Biogen, Celgene, Frequency Therapeutics, GENeuro, Genentech, Merck/EMD Serono, Novartis, Roche, and Sanofi. D.L.A. has an ownership interest in NeuroRx. A.B.-O. is funded by the NIH, ITN, NMSS and MSSOC. A.B.-O. has participated as a speaker in meetings sponsored by and received consulting fees and/or grant support from: Janssen/Actelion; Atara Biotherapeutics, Biogen Idec, Celgene/Receptos, Roche/Genentech, Medimmune, Merck/EMD Serono, Novartis, Sanofi-Genzyme. C.Be. is supported in part by the Bingham Chair in Gastroenterology. He has consulted to or served on advisory boards for Abbvie Canada, Amgen Canada, Bristol Myers Squibb Canada, JAMP Pharmaceuticals, Janssen Canada, Pfizer Canada, Sandoz Canada, Takeda, and has received unrestricted educational grants from Abbvie Canada, Janssen Canada, Pfizer Canada, Bristol Myers Squibb Canada, and Takeda Canada. He has been on the speaker's bureau of Abbvie Canada, Janssen Canada, Pfizer Canada and Takeda Canada. He has received research grants from Abbvie Canada, Amgen Canada, Pfizer Canada, and Sandoz Canada and contract grants from Janssen. R.A.M. receives research funding from: CIHR, Research Manitoba, Multiple Sclerosis Society of Canada, Multiple Sclerosis Scientific Foundation, Crohn's and Colitis Canada, National Multiple Sclerosis Society, the CMSC, the Arthritis

Society and US Department of Defense. She is a co-investigator on a study funded in part by Biogen Idec and Roche Canada. She is supported by the Waugh Family Chair in Multiple Sclerosis. J.O. receives research funding from: Multiple Sclerosis Society of Canada, Multiple Sclerosis Scientific Foundation, and CMSC. G.V.D. is the Chief Bioinformatics Scientist with the National Microbiology Laboratory – Public Health Agency of Canada and has received research support in the last three years from the National MS Society, the Canadian Institute of Health Research, and Genome Canada. E.A.Y. has received research support in the last 3 years from the National MS Society, Canadian Institutes of Health Research, National Institutes of Health, Ontario Institute of Regenerative Medicine, Stem Cell Network, SickKids Foundation, Peterson Foundation, MS Society of Canada, and the MS Scientific Research Foundation. She has received funding for investigator-initiated research from Biogen and has served on scientific advisory boards for Biogen, Alexion and Hoffman-LaRoche. B.B. serves as a consultant to Novartis, UCB, and Roche. B.B. provides non-remunerated advice on clinical trial design to Novartis, Biogen, Teva Neuroscience. B.B. is funded by the NMSS, NIH, and Canadian MS Society. E.W. is funded by the NMSS, NIH, PCORI, DoD and Race to Erase MS. E.W. has received consulting honoraria from Emerald Pharmaceutical, and speaking honoraria from Yoga moves MS, Advanced Curriculum and NeurologyLive. She is a site PI for several pharmaceutical clinical trials (Biogen, Roche, Alexion). H.T. has, in the last five years, received research support from the Canada Research Chair Program, the National Multiple Sclerosis Society, the Canadian Institutes of Health Research, the Multiple Sclerosis Society of Canada, the Multiple Sclerosis Scientific Research Foundation and the EDMUS Foundation ('Fondation EDMUS contre la sclérose en plaques'). In addition, in the last five years, has had travel expenses or registration fees prepaid or reimbursed to present at CME conferences from the Consortium of MS Centers (2018), National MS Society (2018), ECTRIMS/ ACTRIMS (2017-2022), American Academy of Neurology (2019). Speaker honoraria are either declined or donated to an MS charity or to an unrestricted grant for use by H.T.'s research group. All other authors declare no potential conflict of interest.

## Additional information

[1]Department of Medicine (Neurology), The University of British Columbia, Vancouver, BC, Canada. [2]Faculty of Health Sciences, Simon Fraser University, Burnaby, BC, Canada. [3]National Microbiology Laboratory, Public Health Agency of Canada, Winnipeg, MB, Canada. [4]Department of Medical Microbiology and Infectious Diseases, University of Manitoba, Winnipeg, MB, Canada. [5]Curtin School of Population Health, Curtin University, Perth, WA, Australia. [6]Curtin Health Innovation Research Institute, Curtin University, Perth, WA, Australia. [7]Institute for Physical Activity and Nutrition (IPAN), School of Exercise and Nutrition Sciences, Deakin University, Geelong, VIC, Australia. [8]Department of Internal Medicine, Max Rady College of Medicine, Rady Faculty of Health Sciences, University of Manitoba, Winnipeg, MB, Canada. [9]Inflammatory Bowel Disease Clinical and Research Centre, University of Manitoba, Winnipeg, MB, Canada. [10]Department of Community Health Sciences, Max Rady College of Medicine, Rady Faculty of Health Sciences, University of Manitoba, Winnipeg, MB, Canada. [11]Department of Neurology, University of California San Francisco, San Francisco, CA, USA. [12]Department of Pediatrics (Neurology), The Hospital for Sick Children, University of Toronto, Toronto, ON, Canada. [13]Centre for Neuroinflammation and Experimental Therapeutics and Department of Neurology, University of Pennsylvania Perelman School of Medicine, Philadelphia, PA, USA. [14]Department of Neurology, Perelman School of Medicine, University of Pennsylvania, Division of Child Neurology, The Children's Hospital of Philadelphia, Philadelphia, PA, USA. ✉e-mail: helen.tremlett@ubc.ca

