## [Peer Review File · Communications Medicine]

This manuscript has been previously reviewed at another *Nature Portfolio* journal. This document only contains reviewer comments and rebuttal letters for versions considered at *Communications Medicine*.

REVIEWERS' COMMENTS:

Reviewer #4 (Remarks to the Author):

The manuscript explores the relationship between the Mediterranean diet, gut microbiota, and pediatric-onset Multiple Sclerosis. The study identifies various gut microbial taxa that show correlations with both diet and the disease state. The focus is particularly on *Methanobrevibacter*, which exhibits the strongest correlations. However, when analyzing samples provided within 6 days post-FFQ, this strongest association with *Methanobrevibacter* is not observed anymore. The authors attribute this to the low abundance of *Methanobrevibacter* in the sub-group. This finding raises concerns about the cohort size, suggesting it may be too small to establish robust associations and draw meaningful conclusions.

In summary, the manuscript confirms a well-known link between microbiota and diet. While the associations described in the context of Multiple Sclerosis are interesting, the overall novelty of the findings is considered moderate.

Reviewer #4 (Remarks to the Author):

The manuscript explores the relationship between the Mediterranean diet, gut microbiota, and pediatric-onset Multiple Sclerosis. The study identifies various gut microbial taxa that show correlations with both diet and the disease state. The focus is particularly on *Methanobrevibacter*, which exhibits the strongest correlations. However, when analyzing samples provided within 6 days post-FFQ, this strongest association with *Methanobrevibacter* is not observed anymore. The authors attribute this to the low abundance of *Methanobrevibacter* in the sub-group. This finding raises concerns about the cohort size, suggesting it may be too small to establish robust associations and draw meaningful conclusions.

In summary, the manuscript confirms a well-known link between microbiota and diet. While the associations described in the context of Multiple Sclerosis are interesting, the overall novelty of the findings is considered moderate.

We appreciate the reviewer's comment. We would like to clarify that the loss of the association observed in our sub-analysis was attributed to the low *presence*, rather than the abundance, of *Methanobrevibacter*. It is possible that this distinction aligns with the reviewer's intended meaning.

In the sub-analysis of participants within the Diet-microbiota subgroup who provided a stool sample within +/-6 days of the FFQ completion, just over one-third (10/27) of the MS cases were included based on the inclusion criteria: 22 out of 32 controls were also included. In the original analysis of the Diet-microbiota subgroup, *Methanobrevibacter* was present in 33% of MS cases (9/27) compared to a just 3.1% of controls (1/32). In the sub-analysis, the presence of *Methanobrevibacter* was similar, 30% in MS cases (3/10) and 0% in controls. In the sub-analysis, *Methanobrevibacter* was no longer significantly associated with aMED or fiber intake ($P > 0.05$), very likely due to the lower number of participant's samples with *Methanobrevibacter* present.

Since only 3 participant's samples contained *Methanobrevibacter* in the sub-analysis, we do not believe that it is sufficient to draw a robust conclusion on any associations with *Methanobrevibacter* in the small sub-analysis. However, despite the small dataset, our findings for all other taxa were largely consistent with the original analysis.

We modified the manuscript to make it clearer that we meant low prevalence of *Methanobrevibacter*, not low abundance.

Results (page 19):

However, *Methanobrevibacter* was no longer associated with aMED or fiber intake ($P > 0.05$), likely attributable to its limited prevalence as it was detected in only 3 of the 32 remaining participants, all of which had MS.

We have also revised the manuscript to thoroughly discuss the limitation of our sample size.

Discussion (page 25):

Sixth, our modest cohort size limited our ability to draw robust conclusions, especially for specific microbial taxa, such as *Methanobrevibacter*, which was infrequently detected. This size limitation could reduce our ability to uncover true relationships. It also remains possible that some of our findings could have occurred by chance alone. These constraints collectively necessitate a cautious interpretation of the findings. Despite these limitations, this study provides valuable insight into the diet-microbiome-MS relationship. In particular, we identified key potential dietary and microbial risk factors for MS which generate intriguing hypotheses regarding their underlying mechanisms. Furthermore, based on our mediation analysis, we hypothesize that the potential protective effect of a fiber-rich nutritious diet against MS may be attributed to its potential influence on the gut microbiota. To help determine if this hypothesis reflects a causal relationship, conducting a large longitudinal study would be of value.